# Advances in Non-Chemical Tools to Control Poultry Hematophagous Mites

**DOI:** 10.3390/vetsci10100589

**Published:** 2023-09-22

**Authors:** Geralda Gabriele da Silva, Maykelin Fuentes Zaldívar, Lucilene Aparecida Resende Oliveira, Reysla Maria da Silveira Mariano, Daniel Ferreira Lair, Renata Antunes de Souza, Alexsandro Sobreira Galdino, Miguel Angel Chávez-Fumagalli, Denise da Silveira-Lemos, Walderez Ornelas Dutra, Ricardo Nascimento Araújo, Lorena Lopes Ferreira, Rodolfo Cordeiro Giunchetti

**Affiliations:** 1Laboratory of Cell-Cell Interactions, Institute of Biological Sciences, Department of Morphology, Federal University of Minas Gerais, Belo Horizonte 31270-901, MG, Brazil; lucilenearesende@ufmg.com (L.A.R.O.); reyslamariano@rpd.ufmg.br (R.M.d.S.M.); daniellair@ufmg.br (D.F.L.); renata10600@ufmg.br (R.A.d.S.); denise.giunchetti@unifenas.br (D.d.S.-L.); waldutra@icb.ufmg.br (W.O.D.); 2Microorganism Biotechnology Laboratory, Federal University of São João Del-Rei (UFSJ), Campus Centro Oeste, Divinópolis 35501-296, MG, Brazil; asgaldino@ufsj.edu.br; 3Computational Biology and Chemistry Research Group, Vicerrectorado de Investigación, Universidad Católica de Santa María, Urb. San José S/N, Arequipa 04000, Peru; mfumagalli@ufmg.br; 4Laboratory of Hematophagous Arthropods, Department of Parasitology, Federal University of Minas Gerais, Belo Horizonte 31270-901, MG, Brazil; rnaraujo@icb.ufmg.br; 5Laboratory of Ectoparasites, Department of Preventive Veterinary Medicine, Federal University of Minas Gerais, Belo Horizonte 31270-901, MG, Brazil; lorenalopesf@vet.ufmg.br

**Keywords:** ectoparasites, hematophagous, biological, vaccines

## Abstract

**Simple Summary:**

The mites that infest laying hens and broiler chickens in poultry farms have caused great inconvenience to the industry due to the difficulty of controlling or eliminating their populations within the production systems. *Dermanyssus gallinae* and *Ornithonyssus* spp. are the mites that mainly interfere with the health of the poultry, damaging the production and quality of the end product, with special emphasis on *Ornithonyssus sylviarum* and *Ornithonyssus bursa.* The objective of this article is to analyze the impact of hematophagous mites that infest commercial egg and meat production systems and the consequences of this form of parasitism, and discuss the chemical and non-chemical methods of control associated with the use of plants, entomopathogenic fungi, and products based on diatomaceous earth and synthetic silica, and new lines of research aimed at developing vaccines as a new way of controlling these pests.

**Abstract:**

The blood-sucking mites *Dermanyssus gallinae* (“red mite”), *Ornithonyssus sylviarum* (“northern fowl mite”), and *Ornithonyssus bursa* (”tropical fowl mite”) stand out for causing infestations in commercial poultry farms worldwide, resulting in significant economic damage for producers. In addition to changes in production systems that include new concerns for animal welfare, global climate change in recent years has become a major challenge in the spread of ectoparasites around the world. This review includes information regarding the main form of controlling poultry mites through the use of commercially available chemicals. In addition, non-chemical measures against blood-sucking mites were discussed such as extracts and oils from plants and seeds, entomopathogenic fungi, semiochemicals, powder such as diatomaceous earth and silica-based products, and vaccine candidates. The control of poultry mites using chemical methods that are currently used to control or eliminate them are proving to be less effective as mites develop resistance. In contrast, the products based on plant oils and extracts, powders of plant origin, fungi, and new antigens aimed at developing transmission-blocking vaccines against poultry mites provide some encouraging options for the rational control of these ectoparasites.

## 1. Introduction

According to the United Nation’s Food and Agriculture Organization (FAO), the growth in the global poultry sector is driven by a greater demand and purchasing power of the consumer market. Egg production has increased considerably in the past three decades, increasing from 15 million tons in 1961 to 87 million tons in 2020, and chicken meat production increased from 9 million tons in 1961 to 133 million 2020 [1]. The countries with significant numbers associated with chicken production are the United States, China, Brazil, and the European Union [2]. Brazil exports the most chicken meat to the world [3]. China stands out as the largest egg producer, followed by the United States, the European Union, India, and Mexico, with Brazil currently in sixth place [2,3,4].

The blood-sucking mites *Dermanyssus gallinae* (”red mite”), *Ornithonyssus sylviarum* (“northern fowl mite”), and *Ornithonyssus bursa* (”tropical fowl mite”) stand out for causing infestations in commercial poultry farms worldwide [5,6,7,8,9]. These mites directly affect the well-being of animals and consequently cause losses in production that result in significant economic damage for producers [4,6,8,10].

*Dermanyssus gallinae* spends its life cycle hiding in nests, cracks, and crevices and feeds on its host at night. In contrast, *Ornithonyssus* sp. spends its entire cycle on a host, located in the feathers and down around the cloaca region. In cases of severe infestations, *Ornithonyssus* sp. can be found throughout the environment [11,12,13]. The mites are transmitted through direct physical contact between animals, through indirect contact with instruments, cages, and materials infested with the mites, and through wild birds that frequent the laying hens and broilers houses [9,14,15]. These mites can also bite humans, causing ectoparasitic dermatitis [8,16,17].

The elimination or control of these mites has become increasingly difficult due to their resistance to the chemical products available on the market [6,7,17,18,19,20,21,22,23,24,25,26,27,28,29,30,31,32] combined with a change in the system of raising animals in battery cages to that of enriched cages or “cage free” systems that make it difficult to control and observe the animals as they start to live free in sheds and roam freely in pastures [33,34,35]. Due to this scenario, new technologies to control the hematophagous mites without the use of chemical acaricides are needed, such as plant-derived products, entomopathogenic fungi, diatomaceous and silica-based products, semiochemicals, and vaccines. Therefore, this review aimed to compile information regarding *D. gallinae*, *O. sylviarum,* and *O. bursa,* including geographic distribution and the biotechnological advances in the development of new tools against these mites using non-chemical repellent substances.

## 2. Materials and Methods

The data involving the geographic distribution and importance of hematophagous mites as well as chemical control and new technological trends (plant-derived products, entomopathogenic fungi, diatomaceous earth, volatile organic compounds (VOCs), and vaccines) to control them were assessed using the PubMed platform (https://pubmed.ncbi.nlm.nih.gov/. Access from December 2022 to June 2023) and Medical Subject Headings (MeSH terms). Additional information was searched in the Google Scholar database and government websites with data published on the subject. The information was then grouped according to the subject for further analyses: (a) bird mites, (b) poultry mite control methods, (c) biological control of poultry mites, (d) chemical control of poultry mites, and (e) vaccines against poultry mites. The information recovery was used to assess the following: (a) the forms of control that have been currently used against mites present in poultry production farms, (b) the impact caused by infestations of poultry mites, (c) the impact of mite infestation on animal and human health, and (d) the effectiveness of the current control methods and the new perspectives and technologies used or suggested for the control of avian mites.

The review procedure was outlined in topics to maintain the linearity of the information. These topics encompass the following: (a) the mites’ geographically distribution and economic importance in poultry production, (b) the main forms of control that have been used against bird mites and their implications, and (c) new perspectives that have been developed and studied by different research groups that target effective ways to induce mortality or repellency in poultry hematophagous mites.

## 3. Results

### Mites’ Distribution and Their Economic Impact on the Poultry Production System

Hematophagous mites are distributed worldwide and found in poultry farms in various countries (Figure 1). *D. gallinae* is considered the most prevalent mite in poultry farms around the world, with emphasis on European breeding models [29,36,37,38]. It is estimated that approximately 83% of European laying, pullet, and breeder farms, are infested with *D. gallinae* [39,40,41] with losses ranging between USD 130 million and USD 231 million per year [17,22,31,42,43,44,45].

In addition to affecting laying hen farms, the poultry red mite (PRM) is cosmopolitan in range, being able to infest other animals and humans, leading to intensely itchy dermatitis [6,7,37,46] with a zoonotic character [40,46,47,48] and acts as a transmitter vector of disease-causing pathogens [6,22,30,40,44,46,49,50,51,52,53,54]. 

The role of *D. gallinae* as a vector was demonstrated in *Salmonella* spp. [49,55], *Escherichia coli* [41,48], avian poxvirus, and eastern equine encephalitis virus [56]. It can also act as a reservoir for some pathogens, such as *Coxiella burnetii*, *Borrelia afzelii*, *Borrelia burgdorferi* [46], *Erysipelothrix rhusiopathiae* [57], *Mycoplasma gallisepticum*, *Mycoplasma synviae*, *Plasmodium* spp., *Tsukamurella* spp. [47], and *Listeria monocytogenes* [58]. In cases of infestation, these mites cause in the animals a state of anemia, lower feed conversion and weight loss, psychogenic behavior or somatic stress (irritation, pecking, and cannibalism), dermatitis, decreased immunity, and, in extreme cases, death by exsanguination [6,29,31,39,40,59,60]. In addition, mite-infested hens suffer from a reduction in egg laying, and the eggs laid lose their quality (fragile shells and smaller size) [8,29,30,48,51,54,61].

According to Spagarano (2009), the cost of losses due to preventive and control measures against *D. gallinae* are difficult to calculate. The cost in France has been estimated at EUR 4.33/100 poultry and EUR 3.83/100 poultry for cages; in the Netherlands, it ranges from EUR 0.14/poultry to EUR 0.29/poultry [41,49] with productivity losses reaching EUR 0.57 to EUR 2.50 per poultry in one year [62].

*O. sylviarum* is more prevalent in temperate climates, and *O. bursa* is described in tropical and subtropical climates [45] as a causative agent of dermatitis in humans [5,45,63,64,65]. Notably, *O. sylviarum* is the mite primarily responsible for infestations and economic losses involving laying hens and matrix farms in the United States [12,13,20,25,26,66,67,68].

After a severe infestation by hematophagous mites, the poultry becomes anemic, develops dermatitis, exhibits pecking behavior and plucks their feathers, and suffers skin wounds and secondary infections [12,28,38,67]. Poultry infested with *O. sylviarum* also lose weight and have a lower feed conversion ratio, while in laying hens, there is a reduction in egg production and its quality (shell thickness, egg specific gravity, egg weight, and yolk color), resulting in a decrease in the Haugh unit (a measurement of egg protein quality) [13,26].

In a study conducted on a commercial egg farm to evaluate the impact of *O. sylviarum* infestations, Mullens et al. [25] observed a reduction of up to 4% in egg production, along with a reduction in feed conversion and in egg weight of around 0.5% to 2.2%, resulting in estimated losses of 0.70 to 0.10 euros per animal. These numbers, when multiplied by the number of animals from intensive systems, reveal huge losses when associated with all the maintenance costs of the farms. In addition, *O. bursa* is described as the mite primarily responsible for causing infestations and dermatitis in humans [52,64].

The current management of poultry farms for egg production underscores the need to establish new practices related to preventive hygiene [69]. This may be related to the fact that these types of poultry farms are systems where poultry remain for a long period of time [41,70]. It is important to emphasize that raising poultry with systems with a high animal density presents an additional challenge when it comes to controlling poultry hematophagous mites [6,9,11,12,28,71,72].

In the intensive meat production systems, the chickens remain on the farm for 40–50 days [73], and following their slaughter, the environment and all materials and instruments are thoroughly cleaned and sanitized. The short amount of time that the animals remain in the environment, combined with constant cleaning of the facilities and sanitizing of the areas, does not allow mites to reach concentrations to be considered as an infestation: >100,000 mites/poultry for *O. sylviarum* [25,38] and 500,000 mites/poultry for *D. gallinae* [8,39].

Discussions on animal welfare have gained prominence in recent years. The physical spaces in which the animals are kept do not allow them to express their natural behavior [35,38,74,75,76] and consequently impair their health and performance [35,70,76]. Since 2012, the practice of poultry farming in battery cages has been banned in the European Union for ethical reasons and is being replaced by enriched cage systems or “cage free” systems, where animals remain confined, but not in cages [40,41,66,67,77,78,79]. This system makes it difficult to clean the environment, allowing the accumulation of organic matter and creating an ideal environment [66] for mites, such as *D. gallinae* [11], which is reflected in the increase in infestation problems in commercial poultry farms in all of Europe [40].

In addition to changes in production systems, global climate changes in recent years have become a key factor in spreading ectoparasites around the world [80,81], accelerating the development of hemoparasites within the vectors [82]. In general, PRM live well in temperatures ranging from 25 °C to 35 °C and relative humidity from 60% to 80%, so the rise in temperature, combined with changes in wind and rain, contribute to the increased dispersion of mites [83]. The cross-transmission of mites between commercial poultry and wild birds is considered responsible for the dissemination and distribution of the PRM and *Ornithonyssus* spp. [6,50,67].

## 4. Discussion

### 4.1. Chemical Control

The main form of controlling poultry mites is through the use of commercially available chemicals, such as acaricides belonging to the class of organophosphates, pyrethroid, formadin, isoxazolines, carbamate, macrocyclic lactones, and dichloro-diphenyl-trichloroethane (DDT) [11,12,40,51,64,84] (Table 1).

Unfortunately, although these acaricides can be sprayed into the environment, they do not reach some areas, such as crevices, or pass through feathers of the poultry, thus preventing mites from coming into contact with the chemical compounds [31,53,85]. However, the most recent molecule, isoxazoline, can be administered through drinking water [4].

The use of chemicals generates inconvenience for the poultry production industry since most of them are (i) highly toxic, becoming a risk to both the animals and humans; (ii) highly polluting to the environment, leaving residue in water and soil; and (iii) can expose consumers to contaminated eggs and meat [6,7,51,53].

Furthermore, one of the main concerns in using acaricides is the selection of resistant mite populations [7,51,53,86]. These products are used indiscriminately and many producers ignore the legal restrictions in their countries [7,9,19,84], and many of the active compounds are banned in Europe and/or in EU member states [19,50].

**Table 1 vetsci-10-00589-t001:** Efficiency of chemicals tested against poultry hematophagous mites.

Product	Chemical Class	Mite	Test Environment	Mortality *	Action	Reference
Metrifonate (trichlorfon)	Organophosphate	*D. gallinae*	Field	99%	Paralysis and death	[87]
D.D.V.P (Dichlorvos) diluted in water ^1^, D.D.V.P (Dichlorvos) diluted in oil ^2^, deltamethrin ^3^, and amitraz ^4^	Organophosphate ^1^, organophosphate ^2^, pyrethroid ^3^, and formadin ^4^	*D. gallinae, O. sylviarum*	Laboratory	DL50 = 513.34 ppm ^1^, DL50 = 314.15 ppm ^2^, DL50 = 389.57 ppm ^3^, and DL50 = 347.24 ppm ^4,#^	Paralysis and death	[9]
Phoxim 50%	Organophosphate	*D. gallinae*	Field	99%	Paralysis and death	[88]
Cypermethrin and Cypermethrin ^1^ + Chlorpyrifos ^2^	Pyrethroid ^1^ and Pyrethroid ^2^	*O. sylviarum*	Laboratory	>95%	Paralysis and death	[12]
Fluralaner	Isoxazoline	*D. gallinae*	Laboratory	100%	Paralysis and death	[89]
Fluralaner ^1^, Spinosad ^2^, Phoxim ^3^, Propoxur ^4^, Permethrin ^5^, and Deltamethrin ^6^	Isoxazoline ^1^, macrocyclic lactone ^2^, organophosphate ^3^, carbamate ^4^, Pyrethroid ^5^, and Pyrethroid ^6^	*O. sylviarum*	Laboratory	100% ^1^, 98% ^2^, 100% ^3^, 100% ^4^, 12% ^5^, and 14% ^6^	Paralysis and death	[24]
Fluralaner	Isoxazoline	*O. sylviarum*	Laboratory	>90%	Paralysis and death	[20]
Fluralaner	Isoxazoline	*D. gallinae*	Field ^1^ and laboratory ^2^	^1^ 90, 6%, and ^2^ 100%	Paralysis and death	[30]
Phoxim	Organophosphate	*D. gallinae*	Field ^1^ and laboratory ^2^	100% ^1^ and 100% ^2^	Paralysis and death	[30]
Cypermethrin	Pyrethroid	*D. gallinae*	^1^ Field	15.6%	Paralysis and death	[30]
Moxidectin ^1^, ivermectin ^2^, and eprinomectin ^3^	Macrocyclic lactone	*D. gallinae*	Laboratory	45.60% ^1^, 71.32% ^2^, and 100% ^3^	Paralysis and death	[32]
Cypermethrin + Chlorpyrifos + Piperonyl Butoxide ^1^, Alkyl Benzyl Dimethyl Ammonium, Chloride + Glutaraldehyde + Deltamethrin ^2^, Dichlorvos ^3^, and Fluralaner ^4^	Pyrethroid + organophosphosphateus ^1^ + organic compound, pyrethroid ^2^, organophosphate ^3^, and isoxazoline ^4^	*D. gallinae*	Laboratory	^1^ 97.5%, ^2^ 100%, ^3^ 100%, and ^4^ 100%	Paralysis and death	[90]
Fipronil ^1^ and Phoxim ^2^	Fenilpirazóis ^1^ and organophosphate ^2^	*D. gallinae*	Laboratory	77.3% ^1^ and 92.7% ^2^	Paralysis and death	[19]
Ivermectin ^1^, allicin ^2^, Ivermectin + allicin ^3^	Avermectins ^1^ and organosulfur ^2^	*D. gallinae*	Laboratory	100% ^1^, 44% ^2^, and >95% ^3^	Paralysis and death	[84]

* Result considering higher dose and after the end of the last treatment dose; ^#^ 100% mortality dilution of mites. The numbers ^1,2,3,4,5,6^ correspond to product, chemical class, test environment, and mortality.

Fipronil is forbidden to be used in animal used as food [19]. Products belonging to the class of carbamates, organophosphates, and pyrethroids have been banned in the EU for use against *D. gallinae.* The Phoxim is the only veterinary drug registered for the treatment of *D. gallinae* infestations. However, it is not available in all EU countries [49,62]. Moreover, Fluralaner has been widely used to combat ectoparasites, and its use against *D. gallinae* was approved in the EU, but in a restricted way, leading to many producers to resort to and rely on unlicensed chemicals [19].

There is a growing tendency to restrict the use of chemicals, especially in food production systems, to avoid food contamination, exposing workers to potentially harmful substances [19].

### 4.2. Non-Chemical Measures against Blood-Sucking Mites

Extracts and oils from plants and seeds, entomopathogenic fungi, semiochemicals, powder such as diatomaceous earth and silica-based products, and vaccines show promising results and could be investigated as alternatives in addition to being safer methods [22,27,28,53]. Better ways of applying these methods need to be tested in the field, and they could be a part of an integrated management plan that would reduce the use of chemical acaricides [66].

#### 4.2.1. Plant-Derived Compounds

Isolated compounds of plant extracts and essential oils are being studied as possible weapons to be used to combat mites on commercial farms [7,53,91,92]. Some products can be sprayed or administered through contact impregnation in traps (Table 2). These compounds cause mortality or repellence [54]. However, few field assays have been developed. Plant-based products are a promising method because they do not contaminate the environment and can be controlled in doses that do not cause health problems in animals and humans and do not leave residues in food products [6,7,27,51,53].

Under laboratory conditions in which almost all contact between mites and plant-derived oils and extracts is guaranteed, it was possible to obtain mortality rates above 80%, demonstrating the effectiveness of the products [10,23,54,77,97,99,100].

Lundh et al. [94] and Abdel-Ghaffar et al. [1] obtained good efficacy in tests conducted in the field on laying farms using oil and vegetable extracts impregnated in traps. However, for the oil or extract to have an effect on the mite, it needs to come into contact with the product and its effectiveness when used in a trap is limited to captured mites [94]. When used as a spray, it is necessary to develop new methods that ensure the dispersion and maintenance of the product in the environment and in animals for a longer period of time [54], in addition to avoiding the formation of oil films [85].

#### 4.2.2. Entomopathogenic Fungi

These fungi occur naturally in the environment, and as a mechanism of action, they germinate and penetrate the body of arthropods through the cuticle causing paralysis of essential organs resulting in the host’s death [7]. Entomopathogenic fungi are being studied as a tool for use in the control of mites on commercial farms for egg production through spraying in or impregnating traps [7,51,53,85] (Table 3).

It has been demonstrated good results when using entomopathogenic fungi, mainly under laboratory conditions, to control the mite population [51,53,54]. The efficiency of using these fungi to combat bird mites in the field may face some challenges, such as the capacity for fungus proliferation in an uncontrolled risks to animal and human health and does not leave any residue in the end products [7,51,53].

#### 4.2.3. Diatomaceous Earth and Synthetic Silica-Based Products

Both diatomaceous earth (DE) and silica-based products are wettable or in the form of a dry powder that acts on mites with acaricidal efficacy. Their main ingredient, silicon dioxide (SiO_2_), adheres to the mite’s body and immobilizes it leading to desiccation and death [50,102] (Table 4).

The mite remains inside crevices, in organic materials, protected from contact with the products sprayed in the environment [7,53,102]. Ulrich and Han [103] obtained good mortality results against *D. gallinae* with products based on diatomaceous earth under laboratory conditions, but did not obtain significant results when using the same products in field tests, which may have been influenced by the physical properties of the products [103], ambient humidity [78], or changes in climate that did not allow the mites to absorb the products [103].

Although the method does not pose a risk of intoxication [102], silica varies in purity and size, threatening user and animal safety because it irritates the respiratory tract when inhaled [49,62,104].

#### 4.2.4. Semiochemicals

Hematophagous arthropods use chemical (semiochemical) signals to find their hosts and mates. Chemical signals produced by other mites of the same species (pheromones) attract them [105] and chemicals produced by the hosts (allelochemicals) attract (kairomones) or repel them (allomones) [62,104].

Volatile organic compounds (VOCs) were isolated from the uropygial gland of a duck (non-host mite) and can be added directly to chicken feed. After ingestion, chickens release these compounds through the uropygial gland and repel hematophagous mites (*D. gallinae* and *Ornithonyssus* sp.), interrupting their feeding and social behavior [62,106].

Aufray et al. [105] showed that hungry female PRM on a French farm were attracted to a complex mixture of five synthetic VOCs: R-1-octen-3-ol, octanal, nonanal, (E)-2-nonenal, and nonanoic acid, combined in equivolumetric proportions or with (E)-2-nonenal as the main constituent [105]. Cunha (2008) conducted a study on the response of protonymphs to scents from the extracts of conspecific mites and observed that the protonymphs of *D. gallinae* produce a pheromone capable of attracting and stimulating encounters with other protonymphs of *D. gallinae* [16].

Chemical ecology studies involving repellency or attraction in hematophagous mites are still poorly understood, but the results so far indicate a new field for new research and different approaches aimed at developing new products for use against poultry hematophagous mites.

#### 4.2.5. Vaccines

Vaccine research aimed at identifying additional forms of poultry mite control has presented new perspectives in recent years [7,42,51,107] (Table 5), driven by an increase in egg and meat production and consumer demand, and it is performed in an environmentally responsible and organic way [27,96].

It is important to emphasize that vaccines are recognized as solutions that offer protection and can also be used in IPM to cut back on the use of and expenses due to acaricides and the labor involved in spraying them. In addition, vaccines are safe for poultry and do not pollute the environment, leave residues in meat and eggs, or pose a risk of triggering resistance in mites [7,22,27,42,62]. Recent studies have shown vaccines to be effective against ectoparasites, for example, transmissionblocking vaccines (TBVs) [7,27,42,51,107,108,109,110,111,112] (Table 5).

The vaccination process triggers a memory immune response with specific antibody production against the antigenic target, thus providing protection against pathogens that can be controlled by the humoral immune response [47]. TBVs, unlike conventional vaccines, aim to generate a humoral immune response in the vaccinated host, triggering the specific antibody production that is transferred to ectoparasites during blood feeding [18,27,108,109,110,112]. The transferred antibodies act by binding to proteins that are essential for ectoparasite’s survival, disrupting its reproduction, and transmission of pathogens [27,47,111,112,113].

The development of TBVs applied to vector-borne diseases was previously described against the sporozoite forms of *Plasmodium* that cause malaria. The specific antibody production triggered by the recombinant Pfs25 protein acts within the insect vector, interfering in pathogen transmission [108,111]. The antibodies that are transferred during vector blood feeding recognize the gamete surface antigen of *Plasmodium*, affecting the parasite’s life cycle inside the mosquito by preventing its sexual development and interrupting its biological cycle and transmission [18,108,109,111,113]. According to Shimp et al. [113], in the first clinical trial, the experimental sporozoite vaccine reduced the incidence of disease over a 14-month period in about 50% of vaccinated infants and children by preventing subsequent blood-stage infection.

Another example includes the vaccines against *Rhipicephalus microplus*, the cattle tick, TickGARD^®^ in Australia and Gavac^®^ in Cuba, developed from the recombinant Bm86 glycoprotein extracted from the tick’s gut [31,50,112,114]. The mechanism of action is associated with the induction of specific antibodies that target cells in the tick’s gut and either impede their development or cause their death [27,107,112]. Table 5 describes the potential vaccine trials aimed at identifying antigenic targets for blocking the transmission of *D. gallinae*.

**Table 5 vetsci-10-00589-t005:** Antigens used as vaccine candidates against *Dermanyssus gallinae*.

Antigen	Presentation	IgY Levels *	Feeding Challenge/Model	Efficiency/ Mortality	Action	Reference
DGE	Brute	↑ (*p* ≤ 0.05)	in vitro/laboratory	50.60%	Tissue paralysis	[17]
Bm86	Recombinant	↑ (*p* ≤ 0.05)	in vitro/laboratory	23.03%	Interference with the digestive system	[50]
Subolesin	Recombinant	↑ (*p* ≤ 0.05)	in vitro/laboratory	35.10%	Interference in the expression of gene regulation of transcription	[50]
Tropomyosin *D. gallinae* (Der g 10)	Recombinant	↑ (*p* ≤ 0.05)	in vitro/laboratory	19%	Interference with muscle movement and structural integrity of tissue	[31]
Paramyosin (Der g 11)	Recombinant	↑ (*p* ≤ 0.05)	in vitro/laboratory	23%	Interference with muscle movement and structural integrity of tissue	[31]
SME	Brute	↑ (*p* ≤ 0.05)	in vitro/laboratory	78.00%	−	[59]
(Deg-VIT-1) + (Deg-SRP-1) + (Deg-PUF-1)	Recombinant	↑ (*p* ≤ 0.05)	in vivo/Field	0%	−	[59]
PRM	Brute	↑ (*p* ≤ 0.05)	in vitro/laboratory	58.30%	−	[59]
Deg-AKR	Recombinant	↑ (*p* ≤ 0.05)	in vitro/laboratory	42% *	−	[21]
CatD-1 in Montanide™ ISA 71 VG adjuvant	Recombinant	↑ (*p* ≤ 0.05)	in vitro/laboratory	50% *	−	[43]
Dg-CatD-1 DNA	Recombinant	↑ (*p* ≤ 0.05)	in vitro/laboratory	0%	−	[43]
Dg-CatD-1 *E. tenella*	Transgenic	↑ (*p* ≤ 0.05)	in vitro/laboratory	0%	−	[43]
rDg-CatD-1 (Cathepsin D, CatD)	Recombinant	↑ (*p* ≤ 0.05)	in vitro/laboratory	63.40%	Interference in the digestive process	[107]
rDg-CatL-1(Cathepsin L, CatL)	Recombinant	↑ (*p* ≤ 0.05)	in vitro/laboratory	48.01%	Interference in the digestive process	[107]
rDg-Lgm (legumain, Lgm)	Recombinant	↑ (*p* ≤ 0.05)	in vitro/laboratory	18.37%	Interference in the digestive process	[107]
Dg-APMAP	Recombinant	↑ (*p* ≤ 0.05)	in vitro/laboratory	61.88%	Plasma membrane interference	[14]
Deg-CPR-1	Recombinant	↑ (*p* ≤ 0.05)	in vitro/laboratory	>50%	Interference in the digestive process	[115]

* DGE: extract with the mite *D. gallinae*; ↑ (*p* ≤ 0.05): there was an increase in the measured levels of IgY when comparing the control group (non-vaccinated) with the vaccinated group. Bm86: recombinant protein from the tick *R. Microplus*; SME: soluble dust mite extract. Deg-VIT-1: vitellogenin-1; Deg-SRP-1: serpine-1; Deg-PUF -1: protein of unknown function-1; DRP: *D. gallinae* protein.

Harrington et al. [17] used an extract of *D. gallinae* and obtained efficacy of around 50% in a laboratory study. Harrington et al. [17] hypothesized that the feeding chamber interfered with the mite’s feeding, reducing the transfer of antibodies induced by vaccination. Similarly, Xu et al. [107] proposed that their low protection rate could be explained by interference from the feeding chamber, as reported by Harrington (2008).

Harrington et al. [50] used recombinant antigen Bm86 and Subolesin from *R. microplus* against *D. gallinae*, which were able to generate an immune response, but with low efficacy (23.03% and 35.1%, respectively). Bm86 and Subolesin are not found in *D. gallinae*, a fact that may have hampered the development of an effective immune response [50].

Wright et al. [31] and Price et al. [43] extracted proteins from *D. gallinae* macerate to act as possible vaccine candidates but obtained results below 50%. Additionally, Price et al. [43] proposed that the low efficacy may have occurred because of the lack of a specific humoral immune response to the antigen used due to low levels of antigen expression or incorrect folding of the expressed protein.

Bartley et al. [42] obtained a potential vaccine candidate using extracted protein from *D. gallinae* macerate, achieving results of around 50% of protection under laboratory conditions, but it was ineffective when using recombinant proteins and submitting it to a field test. The reason may have been the lack of proper selection of antigen, inducing an inadequate protective immune response [59]. Xu et al. [107], Fujisawa et al. [14], and Murata et al. [115] developed recombinant proteins, rDg-CatD-1, Dg-APMAP-N, and Deg-CPR-1, respectively with an efficacy rate above 50% under laboratory conditions, thus paving the way for new studies.

The transmission-blocking vaccines against poultry mites need to be improved in terms of efficacy, especially for applicability in the field. Poultry farms have large numbers of animals that make administering vaccines time-consuming and costly for the producer. Yet, the expense and time would be justified if poultry farmers had access to effective vaccines that yielded long-term results, making them an advantageous, competitive option compared to the low-cost acaricides currently available on the market [27,111]. In fact, consumers are pressuring the market for sustainable products, particularly food products that replace the use of chemicals with vaccines.

The control methods now in use still show a wide variation in effectiveness, making it difficult to determine their treatment efficacy, especially in the field (Figure 2).

For mites that infest and feed on the feathers and epidermis of poultry, such as *Megninia* spp. [28,72] and *Allopsoroptoides galli* [4,116,117], and for the hematophagous mite *Ornithonyssus* spp., no published references of ongoing research to identify antigens for possible vaccine development were found. With regard to the mites that infest the feathers and epidermis of poultry, antigenic compounds that trigger specific antibodies cannot be transferred to the ectoparasite during the blood meal because the form of feeding does not contain blood, thus limiting the application of TBVs.

### 4.3. Integrated Pest Management (IPM)

Integrated pest management (IPM), or integrated control [28,40], consists of using associated identification, certification, and monitoring techniques to eliminate or control mites on poultry farms [40,78,118]. The IPM requires a study in the environment to better understand the difficulties and choose techniques, methods, or products capable of increasing mite mortality in environments [40,44].

Furthermore, the IPM approach may include chemical control measures using commercial products that are normally sprayed within environments and on animals [31,40,53,85]. In addition, IPM can be used with mechanical or physical measures, including cleaning the environment to remove organic matter and sanitizing the materials and equipment afterward [44]. Importantly, biological controls can be applied through traps impregnated with oils or plant extracts [17,28,40,111].

## 5. Conclusions

Poultry mites present on commercial farms continue to be a problem that needs to be addressed. The accelerated multiplication of mites and the expansion of their geographic distribution due to climate change, associated with the difficulty of developing effective forms of control and their role as a vector, call attention to the role mites play in the poultry industry.

The chemical methods currently used to control or eliminate them are proving to be less effective as they develop resistance. Those chemicals that are still effective can also become less effective over time due to their indiscriminate use. Biological control methods offer many advantages related to animal, human, and food safety, but their efficiency remains low when used in poultry farms, making the need for integrated controls even more essential, but generating higher costs. Moreover, the major limitation of this revision is the restricted information available for mite vaccines that could promote a more effective development in this field.

The development of products based on plant oils and extracts, powders of plant origin, fungi, and new antigens aimed at developing transmission-blocking vaccines against poultry mites so as to provide some encouraging options for the rational control of these ectoparasites.

## Figures and Tables

**Figure 1 vetsci-10-00589-f001:**
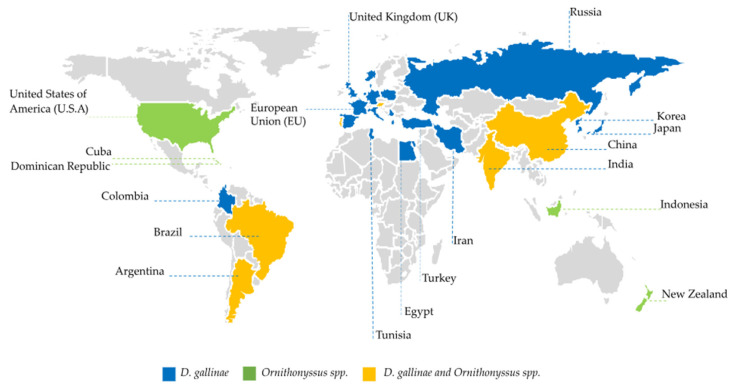
Geographic distribution of *D. gallinae and Ornithonyssus* spp. throughout the world.

**Figure 2 vetsci-10-00589-f002:**
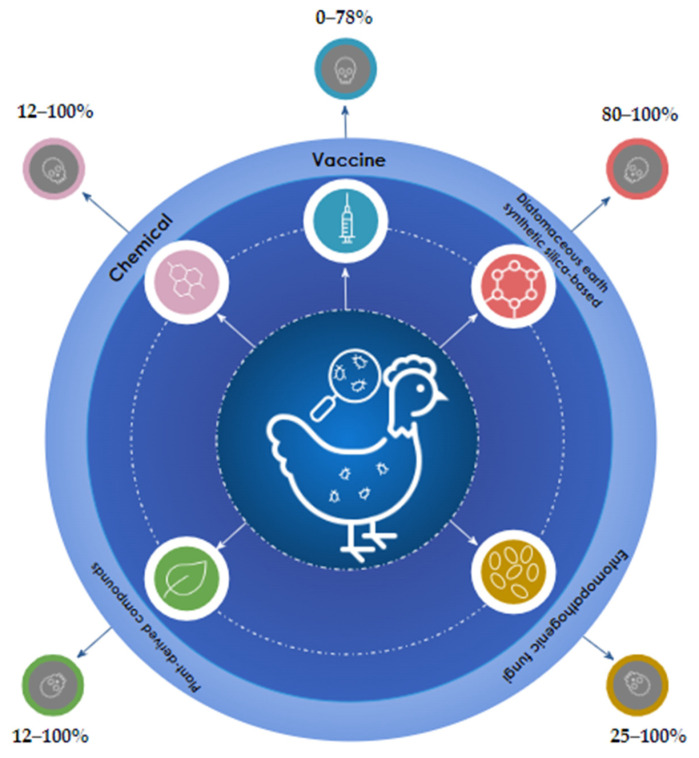
Distinct approaches for controlling mites in laying hens and broilers.

**Table 2 vetsci-10-00589-t002:** Efficiency of plant-derived compounds tested against poultry hematophagous mites.

Product	Mite	Type of Assay	Mortality (M)/Repellency (R)	Action	Reference
*Coffea* aqueous extract ^1^ and *Coffea* chloroform extract ^2^	*D. gallinae*	Laboratory	M = 25% ^1^, and 100% ^2^	Intoxication	[23]
Neem Oil ^1^, Assist ^2^	*D. gallinae, O. sylviarum*	Laboratory	M = 42.86% ^1^, and 15% ^2^	Intoxication	[9]
Oil (individual) of bay, cade, cumin seed, ceylon cardamin, cedarwood, cinnamon, clove bud, clover leaf, coriander, eucalyptus, fir needle, ginger, horseradish, juniper berry, lavender, lemon 10, lemongrass, limedis 5F, mandarin orange, marjoram, mustard, oregano, palmarosa, pennyroyal, peppermint, pimento berry, rosemary, rosemary, peppermint, tea tree, thyme, haiti vetiver, and absinthe	*D. gallinae*	Laboratory	M = 100%	Intoxication	[93]
Basil ^1^ oil or extract, java citronella ^2^, clary sage ^3^, geranium ^4^, nutmeg ^5^, and sage ^6^	*D. gallinae*	Laboratory	M = 56% ^1^, 96% ^2^, 92% ^3^, 93% ^4^, 51% ^5^, and 89% ^6^	Intoxication	[93]
Neem oil	*D. gallinae*	Field	M = 92%	Intoxication	[94]
Neem seed extract	*D. gallinae*	Field	M = 80%	Intoxication	[95]
Eucalyptus essential oil: *Eucalyptus citriodora* ^1^, *E. staigeriana* ^2^, *E. globulus* ^3^, and *E. radiata* ^4^.	*D. gallinae*	Laboratory	M = 85% ^1^, >65% ^2^, 11% ^3^, and 19% ^4^	Intoxication	[77]
2% liquid neem leaf extract + mineral oil + 0.1% degerming agent	*O. sylviarum*	Laboratory	M = > 50%	Intoxication	[86]
Thyme oil	*D. gallinae*	Laboratory	M = 50%	Intoxication	[96]
Lavender oil ^1^, thyme oil ^2^, oregano oil ^3^, and juniper oil ^4^	*D. gallinae*	Laboratory	M => 97% ^1^, 84% ^2^, 50% ^3^, and 50% ^4^	Intoxication	[10]
Acerola cherry oil (individual), bergamot peel, caraway, cinnamon bark, cinnamon leaf, java citronella, clary sage, clove bud, garlic, gurjan balm, hyssop, lavender, lemon peel, lemongrass, lime, marjoram, mint avensis, mustard, onion, pennyroyal, peppermint, pine, rosemary, and white thyme	*D. gallinae*	Laboratory	M = 100%	Intoxication	[97]
Cedarwood oil ^1^, redhead oil ^2^, grapefruit oil ^3^, lemon oil ^4^, peanut ^5^ oil, and sandalwood oil ^6^	*D. gallinae*	Laboratory	M = 48.9% ^1^, 42.2% ^2^, 8.9% ^3^, 33.3% ^4^, 8.9% ^5^, and 20% ^6^	Intoxication	[97]
Clove bud and leaf oil ^1^, steamed lychee oil ^2^, and hemp essential oil ^3^	*D. gallinae*	Laboratory	M = 100% ^1^, 80% ^2^, and 79.26% ^3^	Intoxication	[98]
Ajowan essential oil and ajowan alcoholic extract	*D. gallinae*	Laboratory	>90%	Intoxication	[99]

The authors described general toxic effect on the mites without any association with the system affected. The numbers ^1,2,3,4,5,6^ correspond to product, chemical class, and mortality.

**Table 3 vetsci-10-00589-t003:** Efficiency of entomopathogenic fungi tested against poultry hematophagous mites.

Product	Mite	Test Environment	Mortality *	Action	Reference
Entomopathogenic fungi: *Beauveria bassiana* ^1^ and *Metarhizium anisopliae* ^2^	*D. gallinae*	Laboratory	78% ^1^ and 44% ^2^	Paralysis of essential organs and death	[51]
Solution of entomopathogenic fungi: *Beauveria bassiana* + *Metarhizium anisopliae*	*D. gallinae*	Field	61.7%	Paralysis of essential organs and death	[51]
Fungus Trap: *Trichoderma album*	*D. gallinae*	Field and laboratory	100%	Paralysis of essential organs and death	[53]
Fungus Trap: *Beauveria bassiana*	*D. gallinae*	Field ^1^ and laboratory ^2^	80% ^1^ and 100% ^2^	Paralysis of essential organs and death	[53]
Formulated with entomopathogenic fungi: *Beauveria bassiana*	*D. gallinae*	Laboratory	98%	Paralysis of essential organs and death	[7]
Entomopathogenic fungus: *Aspergillus oryzae*	*D. gallinae*	Laboratory	24.83%	Paralysis of essential organs and death	[101]

* Result considers a higher dose and is obtained after the end of the last treatment dose. The numbers ^1,2^ correspond to product, chemical class, test environment, and mortality.

**Table 4 vetsci-10-00589-t004:** Efficiency of diatomaceous earth and synthetic silica-based products tested against poultry hematophagous mites.

Product	Mite	Test Environment	Mortality	Action	Reference
Neutral detergent 10% ^1^, diatomaceous earth 10% ^2^	*D. gallinae*	Laboratory	100% ^1^ and 97% ^2^	Immobilization, dehydration and death	[7]
Diatomaceous earth 10% ^1^, diatomaceous earth 10% + mechanical cleaning ^2^	*D. gallinae*	Laboratory	93.4% ^1^ and 90% ^2^	Immobilization, dehydration, and death	[100]
Natural diatomaceous earth	*D. gallinae*	Laboratory	100%	Immobilization, dehydration, and death	[103]

The numbers ^1,2^ correspond to product, chemical class, and mortality.

## Data Availability

All data collected were reported in the text.

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
