# Peer review of "Advances in Non-Chemical Tools to Control Poultry Hematophagous Mites"

_vetsci, 2023, doi:10.3390/vetsci10100589_

Round 1
Reviewer 1 Report
My only concern to the authors is to write a few sentences about the limitations of this study in the Conclusion section of your manuscript.
Author Response
Thank you for providing the opportunity to revise and resubmit our manuscript. We acknowledge Reviewer #1 for the comments to enhance its quality. We have made appropriate changes in Revised version - Conclusion Section of the manuscript, that appears as follows:
“(…) Moreover, the major limitation of this revision is the restricted information available of mite vaccines that could promote a more effective development in this field. (…)”
Reviewer 2 Report
Dear Authors,
The idea to give a combined overview for both Dermanyssus and Ornithonyssus is very good, however I have some questions/remarks:
* What about the role of IPM in mite control? Non-chemical control is often included in an IPM approach.
* 3.1: Which parts on Dermanyssus, which on Ornithonyssus? Maintain identical build up and information level for different mites. This is a general remark. It is not always very clear when you focus on Dermanyssus or Ornithonyssus. If no information is available for a specific topic/ mite, please mention this.
* mode of action silica products: please double check in literature.
* See additional remarks in PDF file.
Best regards

Dear authors,
Please have the manuscript reviewed in detail for correct use of language (sentence structure, grammar,..).
Author Response
We are thankful for the Reviewer #2 comments, critical reading and by the opportunity to revise and resubmit our manuscript. We have included an additional topic – “3.4. Integrated pest management (IPM)” – in the revised version of manuscript. In the revised manuscript, the new information appears as follows:
“(…) Integrated pest management (IPM), or integrated control [88][79], consists of using associated identification, certification, and monitoring techniques to eliminate or control mites on poultry farms [37][19][79]. The IPM requires a study in the environment to better understand the difficulties and choose techniques, methods, or products capable of increasing mite mortality in environments [99][79].
Furthermore, the IPM approach may include chemical control measures using commercial products that are normally sprayed within environments and on animals [117][70][53][79]. In addition, IPM can be used with mechanical or physical measures, including cleaning the environment to remove organic matter and sanitizing the materials and equipment afterward [99]. Importantly, biological controls can be applied through traps impregnated with oils or plant extracts [88][56][37][79]. (…)”.
* 3.1: Which parts on Dermanyssus, which on Ornithonyssus? Maintain identical build up and information level for different mites. This is a general remark. It is not always very clear when you focus on Dermanyssus or Ornithonyssus. If no information is available for a specific topic/ mite, please mention this.
We agree with Reviewer #2 with. In fact, we have made a detailed correction throughout the manuscript to include all missing information as requested..
* mode of action silica products: please double check in literature.
We have checked the literature as requested, and the term “intoxication” was changed by “Dehydration and death”.
* See additional remarks in PDF file.
We have made all the requested changes, according the suggestions in the PDF file by Reviewer #2.
Reviewer 3 Report
Very interesting review bringing together the current state of the research in this area and the need for new methods of control.
One point that could be made is that effective control may require an integrated approach, combining several control methods. This point could be added in the conclusions.
Overall the standard of English is very good, however there are several typographical and grammar errors such that the manuscript does require some corrections in the aspect and proof reading.
Author Response
Thank you for providing the opportunity to revise and resubmit our manuscript. We acknowledge Reviewer #3 for the comments to enhance its quality.
As requested, we have included the missing information in the Section “3.4. Integrated pest management (IPM)”
Comments on the Quality of English Language
Overall the standard of English is very good, however there are several typographical and grammar errors such that the manuscript does require some corrections in the aspect and proof reading.
Thanks for the suggestion. We submitted our revised manuscript to a native speaker corrections.
Round 2
Reviewer 2 Report
Dear Authors,
Many thanks for the revised manuscript. In my opinion the quality has improved a lot and gives a good overview of non-chemical tools to control poultry hematophagous mites. The combination of Dermanyssus and Ornithonyssus makes it a very relevant paper.